# The type VI secretion system sheath assembles at the end distal from the membrane anchor

Andrea Vettiger[1], Julius Winter[1], Lin Lin[1] & Marek Basler[1]

The bacterial Type VI secretion system (T6SS) delivers proteins into target cells using fast contraction of a long sheath anchored to the cell envelope and wrapped around an inner Hcp tube associated with the secreted proteins. Mechanisms of sheath assembly and length regulation are unclear. Here we study these processes using spheroplasts formed from ampicillin-treated *Vibrio cholerae*. We show that spheroplasts secrete Hcp and deliver T6SS substrates into neighbouring cells. Imaging of sheath dynamics shows that the sheath length correlates with the diameter of spheroplasts and may reach up to several micrometres. Analysis of sheath assembly after partial photobleaching shows that subunits are exclusively added to the sheath at the end that is distal from the baseplate and cell envelope attachment. We suggest that this mode of assembly is likely common for all phage-like contractile nanomachines, because of the conservation of the structures and connectivity of sheath subunits.

[1] Focal Area Infection Biology, Biozentrum, University of Basel, Klingelbergstrasse 50/70, 4056 Basel, Switzerland. Correspondence and requests for materials should be addressed to M.B. (email: marek.basler@unibas.ch).

Nanomachines related to contractile phage tails share basic components needed for generating mechanical force to puncture target cell membranes and to translocate proteins: a rigid tube with a sharp tip, a contractile sheath and a baseplate[1–4]. The bacterial Type VI secretion systems (T6SS), composed of $\sim13$ core components, are widely distributed among Gram-negative bacteria and deliver various toxins into both eukaryotic and bacterial cells[5,6]. Current model of T6SS mode of action predicts that the assembly starts by formation of membrane complex formed from TssJLM (ref. 7). Next, TssK protein specifically interacts with the membrane complex and likely recruits TssEFG (ref. 8) to form a baseplate together with VgrG/PAAR spike complex associated with T6SS effectors[9–12]. This likely triggers polymerization of Hcp tube and VipA/VipB (TssB/TssC) sheath wrapped around the tube[13–15]. Interestingly, the sheath assembly in *Escherichia coli* is dependent on the presence of TssA forming a dodecameric structure colocalizing with the end of a polymerizing sheath, which is distal from the assembly initiation[16]. However, TssA1 in *Pseudomonas aeruginosa* was shown to interact with TssK1 and TssF1 in the baseplate[17]. This is likely due to the fact that TssAs of *E. coli* and *P. aeruginosa* share little homology and belong to two different subfamilies[17].

Using live-cell fluorescence microscopy, the sheath dynamics can be characterized by three distinct steps: (i) sheath polymerization into a fully extended state, (ii) rapid contraction and (iii) disassembly by ClpV. In several organisms, T6SS sheaths were observed to assemble across the whole cell suggesting that sheath length is limited only by cell diameter. This was shown by direct observations of TssB in *Vibrio cholerae*, *Serratia marcescens*, and *E. coli*[18–20] as well as indirectly by imaging of ClpV localization in *P. aeruginosa*, *Burkholderia thailandensis* and *Bacteroidetes*[21–24]. The sheath contracts to about half of its initial length, presumably propelling the spike and Hcp with the associated effectors into target cells[11,25–29]. The contracted sheath is immediately recognized and disassembled by the AAA+ ATPase ClpV, making VipA/VipB subunits available for the assembly of new structures[21,30–32]. Even though live-cell imaging of T6SS assembly provided unprecedented insights into the mode of action of the T6SS, it is unclear at which end and how new sheath subunits are incorporated into the growing sheath polymer. Interestingly, even for related phage-like contractile nanomachines, there is no direct evidence for the directionality of the sheath assembly[3,33].

Recently, *V. cholerae* was reported to tolerate cell wall synthesis inhibitors by the formation of viable, although non-dividing spherical cells or spheroplasts[34,35]. This allows *V. cholerae* to survive for up to 6 h in the presence of antibiotics 20 times above the minimal inhibitory concentration (MIC) without osmo-protecting agents in the media[34]. Importantly, unlike L-forms of *E. coli* or *Corynebacterium glutamicum*[36,37], *V. cholerae* spheroplasts increase their size significantly as cells continue to grow without cell division[34].

Here we show that the T6SS of *V. cholerae* remains active in cells exposed to ampicillin. Moreover, increased cell size during spheroplast formation correlates with increased sheath length, which allowed us to partially photobleach assembling sheaths and determine at which end of the sheath the soluble subunits are incorporated.

## Results

**Sheath length is limited by cell size.** It was previously reported that in the presence of β-lactam antibiotics, *V. cholerae* forms viable spheroplasts[34]. To test if T6SS remains active in such cells, we exposed exponentially growing *V. cholerae* cells expressing

VipA-msfGFP to ampicillin (500 μg ml$^{-1}$, 100× MIC). In agreement with the published observations, cells quickly lost their rod shape and transformed into spheres by blebbing from the mid-cell while simultaneously losing peptidoglycan (PG) as detected using the fluorescent D-amino acid analogue HADA[34,38,39] (Fig. 1a). Importantly, VipA-msfGFP and ClpV-mCherry2 localization dynamics in spheroplasts suggested that T6SS sheaths cycle between assembly, contraction and disassembly similarly to untreated cells (Fig. 1b,c Supplementary Movies 1–3). Quantification of spheroplast induction and T6SS dynamics revealed that exposure to ampicillin for 40 min at 37 °C generated the highest proportion (89.75%) of spheroplasts displaying dynamic T6SS sheaths (1,007 out of 1,122 cells; Fig. 1d, Supplementary Movie 1). Therefore, this time point was chosen for spheroplast induction in all subsequent experiments unless indicated differently. Due to impaired cell division, the cells increased their size during 40 min exposure to ampicillin from normal rod-shaped cells 1.53 μm$^2$ to spheroplasts 6.59 μm$^2$ (Fig. 1a,e). Interestingly, the average sheath length increased threefold during ampicillin treatment from 0.85 μm to 2.63 μm. Sheaths often spanned across the entire spheroplast and their length correlated ($R^2 = 0.91$) with cell diameter (Fig. 1a,f,g).

Incubation beyond 60 min in the presence of ampicillin (500 μg ml$^{-1}$, 100× MIC) resulted in even larger cells and sheaths, however also occasionally led to outer membrane detachment and cells lysis (Supplementary Fig. 1a). When *V. cholerae* cells were incubated with only 100 μg ml$^{-1}$ (20× MIC) ampicillin, most spheroplasts remained intact for up to 6 h and grew to surface area of up to 28.54 μm$^2$ and assembled sheaths as long as 8.4 μm (Fig. 1h). Overall, these observations indicate that in normal cells the length of T6SS sheaths is limited by available space given by the cell size and when such limitation is absent, the sheaths can assemble to up to ten times longer structures.

**Spheroplasts assemble functional T6SS.** To test if the observed sheath dynamics in ampicillin treated cells corresponds to a functional T6SS, we first monitored Hcp secretion. Spheroplasts were grown to OD = 1, washed twice, inoculated into fresh LB medium supplemented with ampicillin and incubated for 20 min at 37 °C. Proteins from culture supernatant were precipitated and separated by SDS–polyacrylamide gel electrophoresis for immuno-detection of Hcp. This clearly showed that spheroplasts secreted the same amount of Hcp as the untreated rod-shaped cells. Importantly, no Hcp was detected in the supernatant of spheroplasts or rod-shaped cells lacking *vipB*, *tssM* or *tssJ* (Fig. 2a, Supplementary Fig. 2a). Similarly, deletion of *vipB*, *tssM* and *tssJ* decreased number of T6SS sheaths in the spheroplasts to the levels observed in the untreated cells (Fig. 2b).

Furthermore, we monitored protein translocation from spheroplasts into target cells by two different assays: (i) detection of VgrG2 exchange between sister cells and (ii) permeabilization of *E. coli* prey cells. As reported previously for rod-shaped cells[29], spheroplasts formed from cells lacking the tip protein VgrG2 were unable to assemble sheaths (3,500 cells were analysed). When such spheroplasts (Δ*vgrg2*/*vipA-msfGFP*) were mixed with spheroplasts formed from cells with an intact T6SS (*clpV-mCherry2*), on average one sheath assembly was detected in 15 spheroplasts during 5 min (Fig. 2c,d, Supplementary Movie 4). Such frequency of sheath assembly is comparable to what was previously described for intact cells (1 in 20 cells in 5 min)[29]. None of $\sim10,000$ spheroplasts imaged under various conditions contained both msfGFP and mCherry2 signals, indicating that no spheroplast fusion occurred. Furthermore, no sheath assembly was detected in spheroplasts lacking VgrG2,

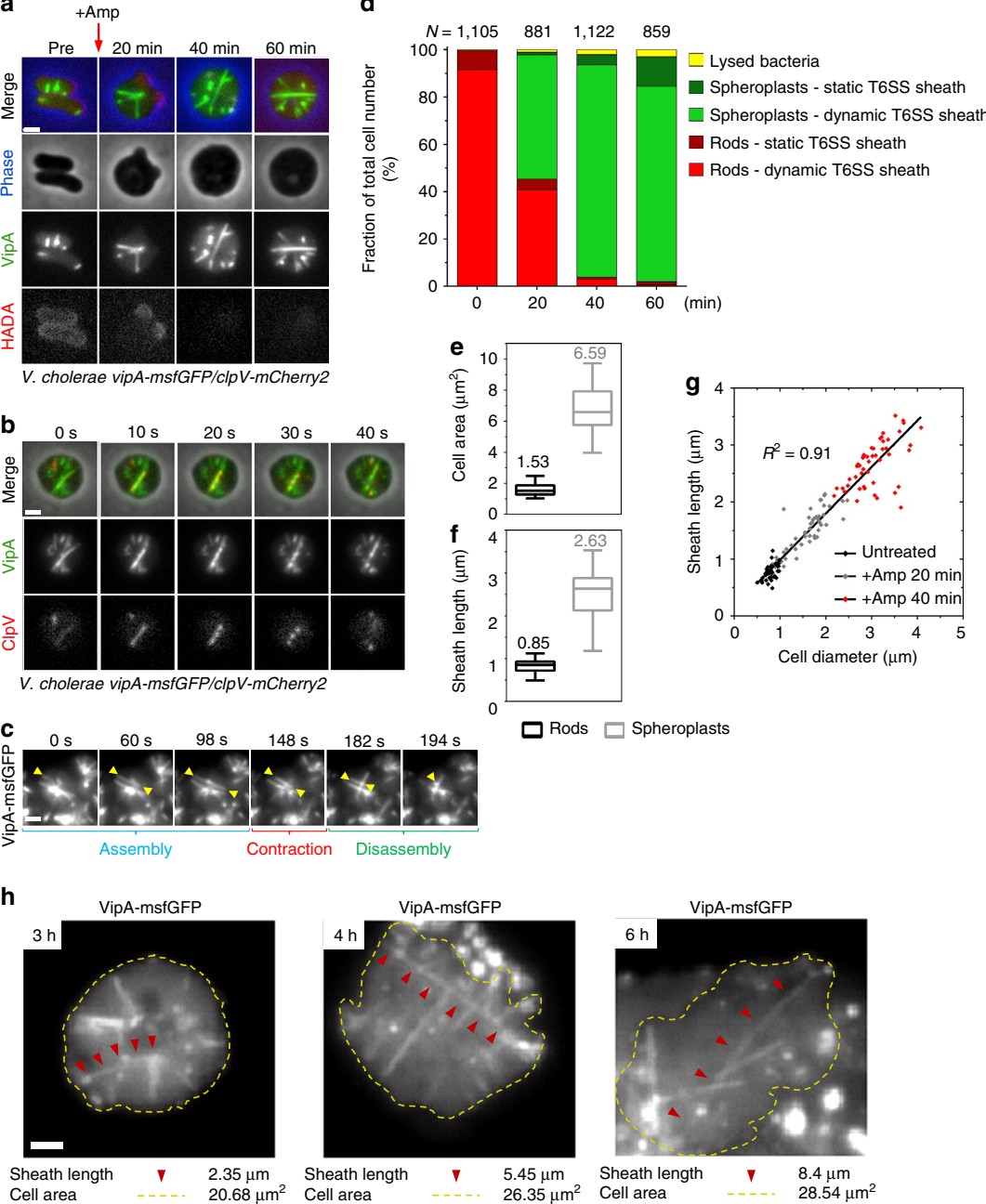

**Figure 1 | *V. cholerae* spheroplasts assemble long and dynamic T6SS sheaths.** (**a**) Cells were grown in the presence of HADA to OD 0.5 before addition of 500 µg ml $^{-1}$ ampicillin (Amp) and harvested for imaging at indicated time points after antibiotic addition. Cell morphology, VipA-msfGFP localizations (sheath dynamics) and HADA staining (PG) were monitored during spheroplast induction. Top row shows a merge of all three channels (phase contrast is pseudo coloured in blue), channels below are displayed individually as grey scale images. Full time-lapse series of each time point are shown in Supplementary Movie 2. Supplementary Movie 3 shows 50 × 50 µm field of view of spheroplasts after incubation with ampicillin for 40 min. Large fields of view for time points 20 min and 40 min are shown in Supplementary Fig. 1b. (**b**) VipA-msfGFP/ClpV-mCherry2-labelled *V. cholerae* spheroplasts (500 µg ml $^{-1}$ ampicillin, 40 min) were monitored for sheath assembly, contraction and disassembly for 5 min. The top row shows a merge of both fluorescence channels and phase contrast, the fluorescence channels are displayed individually as grey scale images. (**c**) VipA-msfGFP labelled spheroplasts (500 µg ml $^{-1}$ ampicillin, 40 min) were imaged for 5 min at a rate of 2 s per frame. Yellow arrow heads indicate a T6SS sheath undergoing assembly into a fully extended state, contraction and disassembly. (**d**) Cells were grown for indicated time points in presence of 500 µg ml $^{-1}$ ampicillin and analysed for cell morphology (lysed, rod shaped and spheroplast) and T6SS dynamics (dynamic or static VipA-msfGFP localization). All data were acquired from three independent experiments. *N*, total number of cells analysed for each time point. (**e,f**) Cell surface area (**e**) and length (**f**) of the longest fully extended VipA-msfGFP structure was measured for 50 rod-shaped cells or spheroplasts formed during 40 min exposure to ampicillin. Data are represented as box-and-whisker plots with minima and maxima; 75% of all data points lay within the box, horizontal lines and numbers represent median values. (**g**) The coefficient of determination ($R^2$) between sheath length and cell diameter was calculated from 50 cells in each category. Data are fitted with a linear regression line. All data were acquired from three independent experiments. (**h**) Spheroplasts formation was induced for the indicated time by incubation of cells at 37 °C on 1% agarose pads containing 100 µg ml $^{-1}$ ampicillin. The VipA-msfGFP fluorescence channel is displayed. Cell outlines are shown in yellow, red arrow heads indicate the longest sheath within the observed cell. Scale bars, 1 µm.

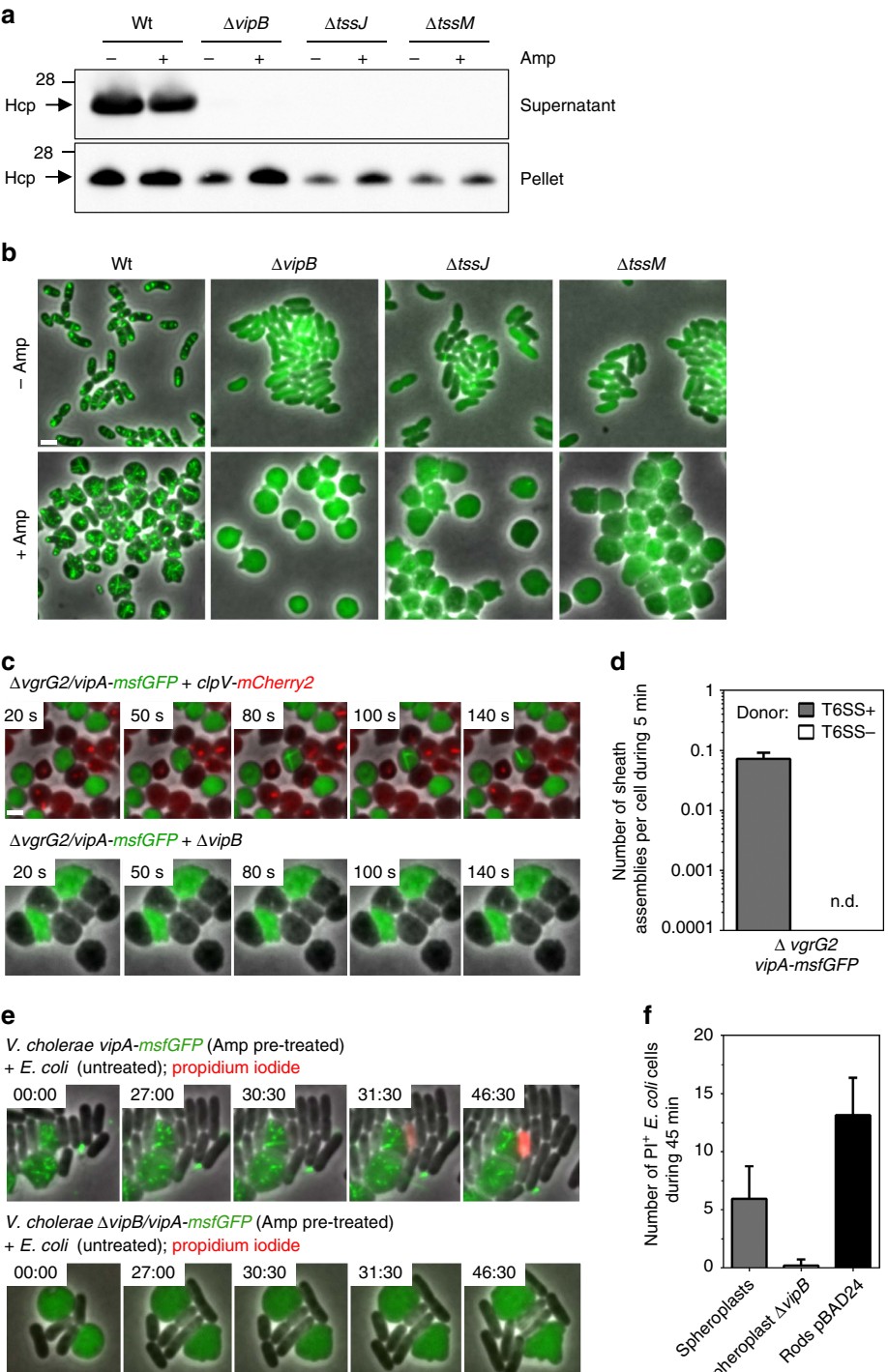

**Figure 2 | The T6SS appears to be functional in ampicillin-induced spheroplasts.** (**a**) Preinduced spheroplasts (500 μg ml $^{-1}$ ampicillin for 40 min) and untreated rod-shaped cells were washed twice and inoculated in fresh medium. Presence of Hcp was detected in culture supernatant and cell pellets of indicated strains after 20 min incubation. The molecular weight is indicated on the left in kilodaltons. Full blots and Coomassie Blue stained gels are provided in Supplementary Fig. 2a. (**b**) Indicated strains (all *vipA-msfGFP* background) were monitored for sheath assembly. Representative images of untreated rod-shaped cells and spheroplasts (500 μg ml $^{-1}$ ampicillin for 40 min) are shown. Scale bar, 2 μm. (**c,d**) VipA-msfGFP labelled recipient spheroplasts (ampicillin 500 μg ml $^{-1}$, 40 min) lacking VgrG2 were co-incubated with either ClpV-mCherry2 labelled (top) or T6SS-negative (Δ*vipB*; bottom) donor strains. Sheath assembly in recipient cells was monitored (**c**) and quantified (**d**) in N = 2,000 VipA-msfGFP cells for each mixture. See Supplementary Movie 4 for complete time-lapse series. Full fields of view are provided in Supplementary Fig. 2b. (**e**) VipA-msfGFP labelled T6SS positive (top) and T6SS-negative (bottom) *V. cholerae* spheroplasts (pretreated in ampicillin 500 μg ml $^{-1}$ for 40 min, washed in LB) were co-incubated with MG1655 prey cells (grey) on agarose pads containing 100 μg ml $^{-1}$ ampicillin and 1 μg ml $^{-1}$ PI. (**f**) Number of PI $^+$ *E. coli* cells was quantified from 20 fields of view (30 × 30 μm) during 45 min co-incubation with indicated *V. cholerae* strains. See Supplementary Movie 5 for complete time-lapse series. Full fields of view are provided in Supplementary Fig. 2c. All data were acquired from three independent biological experiments and are represented as mean ± s.d. Scale bar, 1 μm.

which were co-incubated with spheroplasts formed from a secretion incompetent ΔvipB strain (Fig. 2c,d).

To test if spheroplasts deliver toxins into prey cells and lyse them, we mixed ampicillin-induced *V. cholerae* spheroplasts with ampicillin resistant *E. coli* (carrying pUC19) at 1:1 ratio on agarose pads containing $100\,\mu g\,ml^{-1}$ ampicillin as well as propidium iodide (PI) to identify permeabilized cells. After 45 min of co-incubation with T6SS positive spheroplasts, we detected on average 5.95 PI-positive *E. coli* cells (out of on average 85 *E. coli* cells in total) in a $30 \times 30\,\mu m$ field of view with at least 50% confluence. This was comparable to average 13.15 PI positive *E. coli* cells detected upon co-incubation with rod-shaped *V. cholerae* cells. Importantly, no PI-positive *E. coli* cells were detected in a mixture with T6SS-negative spheroplasts (Fig. 2e,f; Supplementary Movie 5). Overall, these data support that the T6SS is functional in ampicillin-induced *V. cholerae* spheroplasts, suggesting that PG crosslinking is at least partially dispensable for T6SS activity.

**Sheaths quickly contract to half their length.** Measurement of sheath length in rod-shaped cells is potentially imprecise because it is limited by spatial resolution of an optical microscope. To provide a better estimate for an extent to which sheaths contract, we measured the level of sheath contraction in spheroplasts. Dynamic processes such as sheath polymerization and contraction can be visualized in a kymogram by plotting fluorescence signal as function of distance over time along a designated line profile. This allows to distinguish between distinct steps of sheath dynamics in a single image. We followed 50 sheaths, which transitioned from an extended to a contracted state within two consecutive frames at a frame rate of $2\,s\,frame^{-1}$. On average, the comparison of the length of extended and contracted sheaths shows that the sheaths contract to 48.7% of the extended sheath length (Fig. 3, Supplementary Fig. 3).

In addition, to measure speed of contraction, we imaged spheroplasts for 5 s with a frame rate of 500 frames per sec and identified contractions by image analysis (Fig. 3a, Supplementary Fig. 3). Interestingly, in all 5 cases the sheaths contracted between two consecutive frames, therefore faster than in 2 ms (Fig. 3b). The longest observed contracting sheath was $3.29\,\mu m$ long and

contracted by $1.6\,\mu m$ to $1.69\,\mu m$ (Fig. 3a,b). Therefore, we can estimate that sheath contraction occurs faster than $800\,nm\,ms^{-1}$.

**Sheath polymerizes at the distal end.** Live-cell fluorescence microscopy shows that sheath assembly starts from one site in a cell and in time progresses across the whole width of the cell. The initial point of assembly is static, suggesting that this is where the sheath is connected to the membrane anchored baseplate[7,16]. In theory, there are three possible mechanisms how sheath polymers may assemble from soluble subunits: (i) subunits are added at the end distal to the baseplate, (ii) subunits are inserted at the baseplate or (iii) subunits are inserted in between the existing subunits along the whole polymer. To distinguish the three alternative mechanisms, we reasoned that we could photobleach a section of an assembling sheath and then monitor intensity and localization of the photobleached section relative to the rest of the assembling sheath. Depending on the three theoretical mechanisms of sheath assembly described above, these results may be expected: (i) photobleached section of the sheath would be fixed in intensity and relative localization, (ii) photobleached section would have fixed intensity and move away from the site of assembly initiation and (iii) photobleached section would change intensity, localization and size (see the three possible mechanisms in Supplementary Fig. 4).

With our experimental set-up, we estimated the photobleached area to be $\sim 0.8\,\mu m$ in diameter, which is an average length of a sheath in a rod shape cell and we were therefore unable to bleach polymerizing sheaths without photobleaching most of the cell (Supplementary Fig. 5). Spheroplasts assemble sheaths that are several times longer therefore such sheaths should be possible to photobleach only partially. However, wild-type spheroplasts assemble multiple sheaths, which are often difficult to resolve and individually track using standard wide field microscopy (Fig. 1d).

Interestingly, the *V. cholerae* strain lacking vgrG1 and vasX was reported to contain about three times less sheaths than wild-type cells, while displaying normal T6SS dynamics (Fig. 4a)[29]. Indeed, deletion of vgrG1 and vasX decreased number of sheaths assembled per cell in 2 min from 4.44 in wild-type rod-shaped cells to 1.57 in double mutant cells and from 7.08 to 2.53 in the respective spheroplasts (Fig. 4b). Importantly, the sheaths often

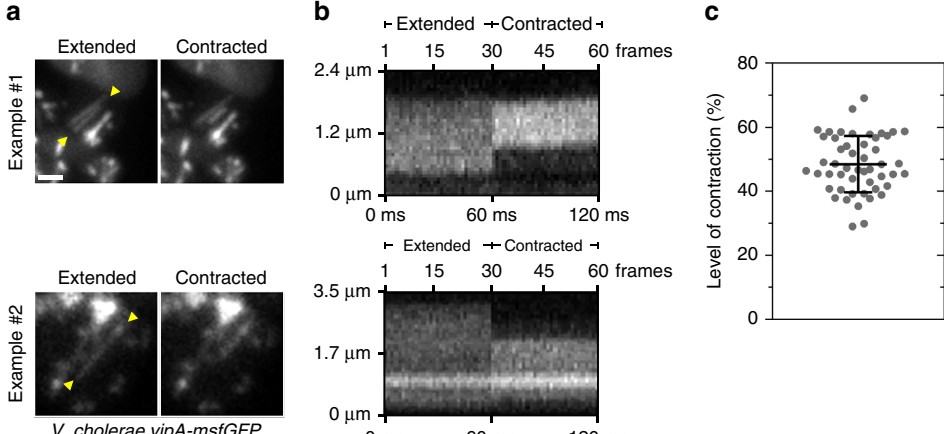

**Figure 3 | The sheath contracts within $<2\,ms$ to half its extended length.** (**a**) High-speed imaging (500 frames per sec) of two representative contraction events in VipA-msfGFP labelled spheroplasts (ampicillin $500\,\mu g\,ml^{-1}$, 40 min). Depicted are maximum intensity projections of 30 frames pre and post sheath contraction. In between the yellow arrow heads a line profile was drawn for subsequent kymogram analysis shown in **b**. Additional examples can be found in Supplementary Fig. 3. Scale bar, $1\,\mu m$. (**c**) Level of sheath contraction was measured for 50 individual contraction events in spheroplasts imaged at a rate of 2 s per frame. Contracted sheath length was subtracted from extended state and normalized to extended sheath length. Data were acquired from three independent experiments and are represented as mean ± s.d.

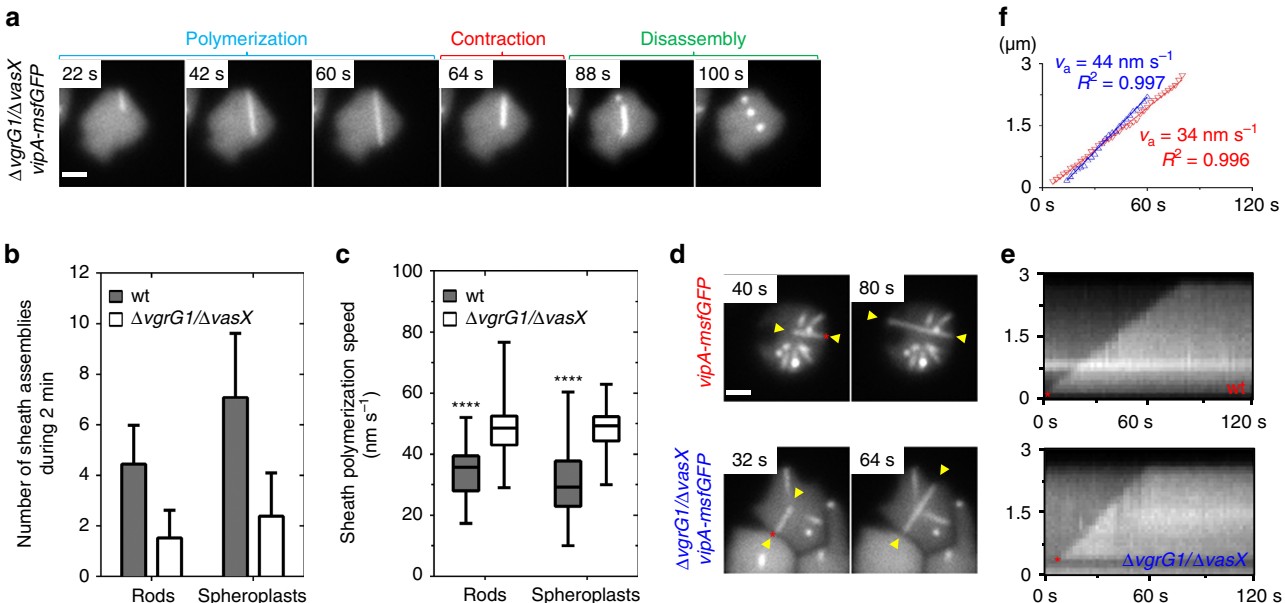

**Figure 4 | Sheath polymerization speed negatively correlates with number of sheaths per cell.** (**a**) VipA-msfGFP labelled $\Delta vgrG1/\Delta vasX$ spheroplasts (500 µg ml$^{-1}$ ampicillin, 40 min) were imaged for 5 min at an acquisition rate of 2 s per frame. Different steps of T6SS sheath dynamics are labelled. (**b**) Number of sheath assemblies (VipA-msfGFP labelled) was assessed in rod-shaped cells and spheroplasts. For each strain and condition, 50 cells were analysed. Data are represented as mean ± s.d. (**c**) Polymerization speed was calculated from kymograms of 50 sheaths for each strain and condition. Data are represented as box-and-whisker plots with minima and maxima; 75% of all data points lay within the box, horizontal line and numbers represent median values. Two-way ANOVA; ****$P < 0.0001$. (**d–f**) Indicated strains were monitored for sheath assembly for 2 min. Examples of sheath polymerization are shown (**d**). Corresponding kymograms (**e**) and corresponding sheath polymerization speed measurements (**f**) ($v_a$) of wt (red) and $\Delta vgrG1/\Delta vasX$ (blue) spheroplasts are shown on the right. Linear regression models were fitted and coefficient of determination ($R^2$) was calculated. Red asterisks indicate the origin of sheath polymerization. Yellow arrow heads indicate polymerizing sheath and were used to draw a straight line for kymogram analyses on the right. Scale bars, 1 µm.

assembled into structures extending across the whole spheroplast (Fig. 4a,d). Based on kymogram analysis, we also noticed that sheath assembly speed increased significantly (two-way analysis of variance (ANOVA), $P < 0.0001$) from 38 nm s$^{-1}$ in wild-type cells to 55 nm s$^{-1}$ in $\Delta vgrG1/\Delta vasX$ cells (Fig. 4c,e,f). However, no significant differences in polymerization speed were found between rod-shaped cells and spheroplasts (two-way ANOVA, $P < 0.58$; Fig. 4c). In all cases sheath polymerization speed was constant over time ($R^2 > 0.99$; Fig. 4f). This suggests that the absence of cross-linked PG and increased sheath length has no impact on sheath assembly kinetics.

To address sheath assembly mechanism, VipA-msfGFP localization was monitored in $\Delta vgrG1/\Delta vasX/vipA-msfGFP$ spheroplasts for 2 min at a frame rate of 2 s per frame (Fig. 5a–d, Supplementary Movie 6). After 30 s (15 frames), cells were partially photobleached using a laser beam following a straight line (Fig. 5a,d). Since the subcellular localization and timing of sheath assembly is random for each individual cell and happens very quickly, we performed the photobleaching randomly across the whole field of view. Later, we analysed the collected time-lapse series and manually searched for sheaths that: (i) were polymerizing during the time before photobleaching, (ii) were only partially photobleached and (iii) kept polymerizing in the same direction after photobleaching. For 25 of such events, we plotted fluorescence intensity along the assembling sheath structure in time (kymogram) (Fig. 5b) and analysed speed of sheath assembly (Fig. 5c) as well as intensity and localization of the photobleached section relative to the point of the sheath assembly initiation (Fig. 5e–g). In all cases, the sheaths polymerized along a straight line and sheath polymerization speed before photobleaching was identical to that after photobleaching as determined by linear regression analysis ($R^2 > 0.99$).

This provides a strong evidence that indeed the same sheath assembly was observed during the whole time lapse (Fig. 5c, Supplementary Fig. 6). Furthermore, we identified few sheaths which contracted after photobleaching (Supplementary Fig. 6), indicating that the photobleaching has no influence on sheath dynamics.

Importantly, the non-bleached section of the sheath assembled before photobleaching as well as the section that was photobleached remained at the same distance from the point of assembly initiation (Fig. 5f). At the same time, low intensity sheath structures extended at the end opposite of the assembly initiation. This section of the sheath was presumably assembled from the partially photobleached subunits that were present in the cell cytosol during laser illumination (Fig. 5e,f). Furthermore, in all cases, the photobleached sections of sheaths retained a fluorescence intensity similar to the intensity of the background cytosolic fluorescence and no recovery of the fluorescence signal was observed (Fig. 5g). Overall, these observations support that sheath subunits are added at the end distal to the baseplate and are not incorporated at the end close to the baseplate or between the subunits of an existing sheath polymer.

## Discussion

The remarkable ability of the *V. cholerae* cells to form large viable spheroplasts with functional T6SS allowed us to image assembly of partially photobleached sheaths and show that sheath subunits are added at the end distal from the assembly initiation. The first ring of a sheath assembles on a baseplate and the next sheath rings assemble on the previous sheath rings. Since T6SS sheaths share common fold and inter-subunit connectivity with phage-like sheaths of other contractile nanomachines[3,40–42], we suggest

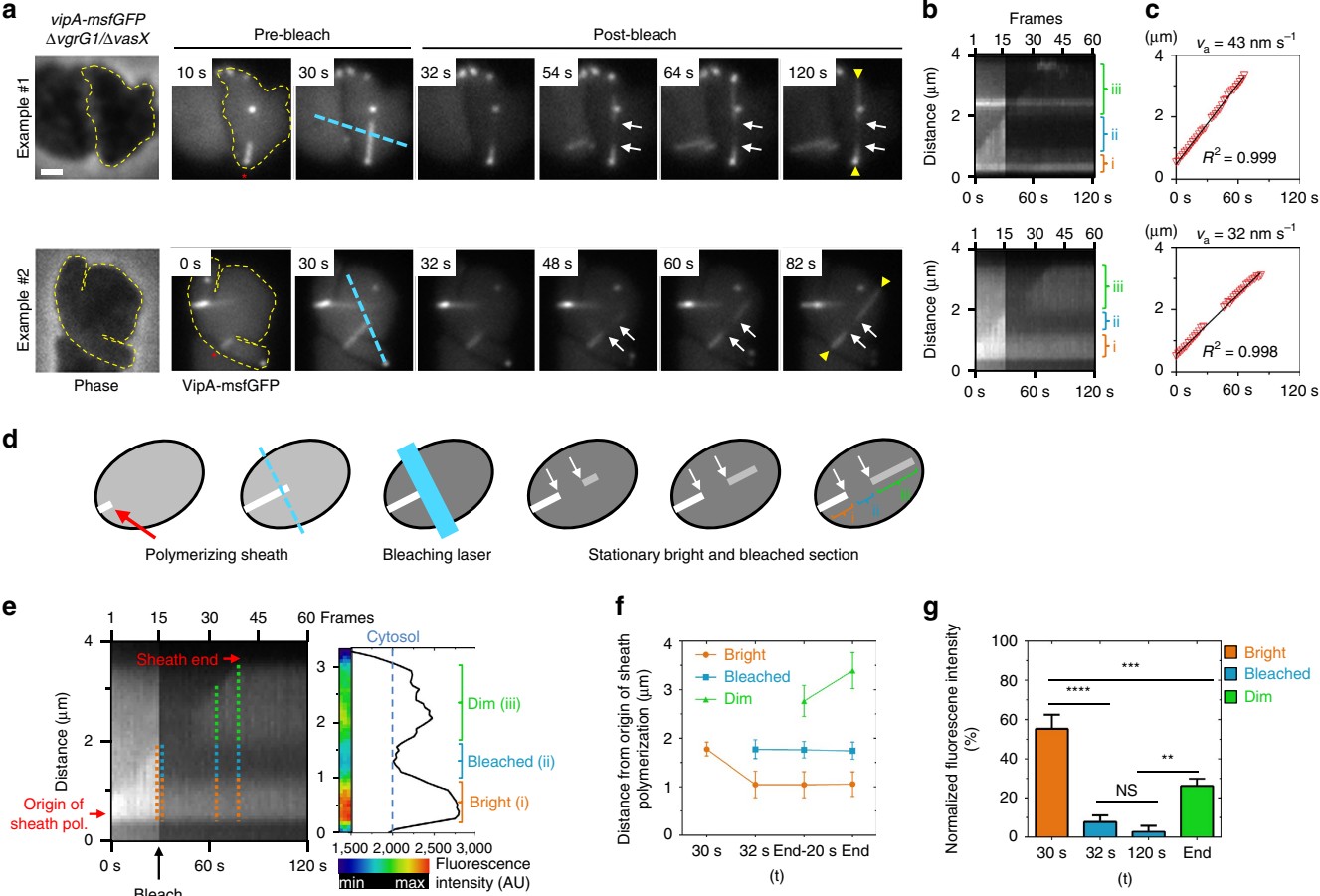

**Figure 5 | Sheath subunits are incorporated at the end distal from the assembly initiation.** (**a**) Phase contrast images of VipA-msfGFP labelled Δ*vgrG1/* Δ*vasX* spheroplasts (500 µg ml$^{-1}$ ampicillin, 40 min) are shown on the left. Cell shapes are outlined by dashed yellow lines based on the phase contrast image. Spheroplasts were monitored for sheath assembly for 2 min at a rate of 2 s per frame. After 30 s of image acquisition, bleaching laser was passed along the indicated line path (dashed turquoise line). Red asterisks indicate the origin of sheath polymerization. White arrows indicate bleached section on the sheath. In between the yellow arrow heads a straight line was drawn for subsequent kymogram analyses and fluorescence intensity measurements (**b**) as well as for the determination of sheath polymerization speed ($v_a$) (**c**) A linear regression model was fitted and coefficient of determination ($R^2$) was calculated. Additional examples as well as corresponding time-lapse movies can be found in Supplementary Fig. 6 and Supplementary Movie 6. Scale bar, 1 µm. (**d**) Conceptual representation of results obtained from photo bleaching experiments. (**e**) Detailed analysis of the example #2 in **a**: fluorescence intensity measurements and heat map were generated using Fiji. Based on fluorescence intensity curves, different sections on sheath were classified as 'bright' (i, orange), 'bleached' (ii, turquoise) and 'dim' (iii, light green). (**f**) For each section, corresponding distances from the origin of sheath polymerization was measured as shown by dotted lines on the kymogram in **e**. The first two distance measurements were performed directly before and after photobleaching. The latter two measurements were performed 20 s before the end of sheath polymerization and when sheath polymerization was completed. (**g**) Fluorescence intensity was determined from all 25 line profiles of successfully photobleached sheaths as shown in **e**. Fluorescence intensity of the corresponding section on the sheath was measured at the indicated time points and was normalized for each time point to the cytosolic background fluorescence (defined as 0% fluorescence intensity). Data are represented as mean ± s.e.m. One-way ANOVA; ****$P < 0.0001$, ***$P < 0.001$, **$P < 0.01$, NS, non-significant.

that the same assembly mechanism likely applies to all related contractile tails. This mode of assembly was indeed predicted based on available structures of baseplates and sheaths[3,43] because sheaths strongly attach to baseplates through conserved linkers[14,15,42]. This mode of sheath assembly has important implications for the role of certain TssA proteins in T6SS assembly. The distal end of T6SS sheath in *E. coli* is capped by TssA, which is required for sheath polymerization, interacts with Hcp and TssC, and colocalizes with the distal end of a polymerizing sheath[16]. Our data therefore confirm the previous suggestions that proteins similar to TssA in *E. coli* promote insertion of new sheath and tube subunits[16]. However, since TssA proteins vary in their predicted structure and some T6SS clusters seem to lack TssA proteins similar to TssA in *E. coli*[17], it is therefore likely that certain sheath-tube complexes assemble

without a need for TssA-like cap or that another, yet to be identified protein, fulfills this role.

Ampicillin treatment of *V. cholerae* cells seems to have no obvious effect on T6SS function. The sheaths in spheroplasts undergo assembly, contraction and disassembly by ClpV like sheaths of untreated cells (Fig. 1b,c)[18–21]. Additionally, spheroplasts secrete Hcp, deliver VgrG2 to neighbouring cells and kill target cells (Fig. 2a,c,e). This suggests that T6SS is functional without strong anchoring of the membrane complex to an intact peptidoglycan. Indeed, peptidoglycan has to be cleaved locally to allow for T6SS assembly[44]. Importantly, the spheroplasts have intact inner and outer membranes[35] (Supplementary Fig. 1a) and thus the membrane complex embedded in both membranes[7] likely provides strong enough attachment to allow for T6SS function. Indeed, the sheath

assembly in spheroplasts depends on the presence of both TssM and TssJ (Fig. 2b).

The sheath-tube polymerization in spheroplasts may progress for up to several micrometres, and the sheath length seems to be only limited by the physical space available between the opposing sides of the cell (Fig. 1g). Cells with fewer assembled sheaths, thus presumably with more available soluble subunits, assemble sheaths faster than the cells with many assembled sheaths (Fig. 4c). This suggests that the speed of sheath polymerization may be partially limited by the amount of available soluble subunits. Furthermore, when the amount of sheath or tube subunits is below a certain threshold, only short dynamic sheaths assemble[18,29].

The fact that T6SS sheath length appears to be unregulated (Fig. 1g,h) stands in contrast with the strictly regulated assembly mechanism of most of the related phage-like contractile tails, where a tape measure protein is critical for tail assembly and its length defines the number of tube and sheath rings assembled into the particle[3,43]. Interestingly, shorter T6SS sheaths were suggested to be sufficient for effector delivery by *V. cholerae*[29], however, longer sheaths may increase the efficiency of effector delivery. Longer sheaths would also likely deliver larger amounts of Hcp associated effectors during a single sheath contraction[28]. Overall energy released during sheath contraction is likely proportional to the sheath length[42], therefore long sheaths might also be required for delivery of large folded hydrophilic effectors associated with the spike complex[11,45]. On the other hand, it is possible that after reaching a certain threshold sheath length, the energy of contraction would be greater than the energy needed to destabilize T6SS baseplate or membrane complex and such sheaths would fail to deliver any effectors.

Overall, we show that imaging of large viable spheroplasts provides new opportunities to dissect T6SS assembly, function and dynamics. More generally, large cells lacking cell wall may be used to study systems, which are challenging to visualize by light microscopy in small bacterial cells.

## Methods

**Strains.** Parental *V. cholerae* 2740-80 and VipA-msfGFP labelled derivatives were described previously[15,18,21,29]. *E. coli* MG1655 were used as prey cells for cell permeability assays. A detailed strain list can be found in Supplementary Table 1. Bacteria were grown in Luria–Bertani (LB) broth at 37 °C. Liquid cultures were grown aerobically. Antibiotic concentrations used were streptomycin ($100\,\mu g\,ml^{-1}$) and ampicillin ($100–500\,\mu g\,ml^{-1}$).

**Spheroplast induction.** Similar procedures as described recently were applied[34]. Briefly, *V. cholerae* overnight cultures were diluted 1:1,000 into fresh LB and grown until early exponential growth phase (OD ∼0.5) before the addition of ampicillin ($500\,\mu g\,ml^{-1}$, 100× MIC). If not indicated differently, spheroplast induction cultures were incubated for 40 min at 37 °C, 100 r.p.m. and subsequently harvested by centrifugation (2 min, 3,000*g*) for further experimental procedures. For incubation times beyond 1 h, $100\,\mu g\,ml^{-1}$ (20× MIC) ampicillin was used. Cell diameter and sheath length of the longest structure per cell during the observation period were measured manually using 'straight line' tool in Fiji[46].

**HADA staining.** For monitoring PG alterations during spheroplast induction blue fluorescent D-amino acid analogue HADA was used[38]. HADA (50 mM) was added to the cultures at OD ∼0.2. In *V. cholerae*, D-amino acids analogues are incorporated by penicillin-insensitive L,D transpeptidases allowing to stain PG in the presence of cell wall targeting antibiotics[34]. Before imaging, cells were washed 2x in fresh LB to get rid of excess dye.

**Hcp secretion assay.** Briefly, cells were grown to OD = 0.8–1.2. Spheroplasts were induced as described above. Subsequently, 1 ml of rod-shaped cells and spheroplasts were harvested by centrifugation (3,000*g*, 2 min) and washed twice before dilution into 1 ml of fresh medium supplemented with ampicillin ($500\,\mu g\,ml^{-1}$) for spheroplasts cultures. Cells were incubated while shaking for 20 min, 100 r.p.m., 37 °C. Bacterial pellets and supernatants were separated by centrifugation (3,000*g*, 2 min). For detection of secreted Hcp in culture supernatant, 900 µl of supernatant were concentrated by TCA/acetone precipitation[47]. For detection of Hcp in cell

pellets 250 µl cells were harvested and resuspended in 80 µl Laemmli buffer and boiled for 95 °C. Proteins were separated on Novex 4–12% Bis-Tris SDS–polyacrylamide gel electrophoresis gels (Thermo Fisher Scientific) and transferred to nitrocellulose membrane for immuno-detection as previously described[29] or proteins were visualized directly by Coomassie Blue staining.

**Interbacterial protein complementation assay.** Similar procedures were applied as described previously[29]. Overnight cultures were washed once in LB and diluted 1:100 into fresh medium and cultivated to an OD at 600 nm of 0.5. Spheroplasts were induced as described above. Cells from 1 ml of the culture were concentrated to OD 10, mixed at a ratio of 1:4 (recipient to donor), subsequently spotted on a thin pad of 1% agarose in LB and covered with a glass coverslip. Spheroplasts were immediately imaged during an observation period of 1 h in multiple 5 min time-lapse series. For image analysis, total number of green fluorescent protein (GFP) positive cells were counted using the built in 'find maxima' function in Fiji with a 'noise tolerance' setting of 250 and activated edge maxima exclusion. For quantification of the number of sheath assemblies in recipient cells from time-lapse movies the 'temporal colour code' function was used. Three independent biological replicates were analysed.

**Cell permeability assay.** *E. coli* MG1655 prey cells, transformed with empty pUC19 vector (Thermo Fisher Scientific) mediating ampicillin resistance, were grown in the presence of ampicillin ($100\,\mu g\,ml^{-1}$) to OD 1. Simultaneously, *V. cholerae* spheroplasts and rod-shaped control cells harbouring the pBAD24 vector mediating ampicillin resistance (predators) were grown and induced as described above. For both prey and predator cells, 1 ml of the culture was harvested and concentrated to OD 10. Cells were mixed at a ratio of 1:1 (prey to predator) and subsequently spotted on a thin pad of 1% agarose in LB containing ampicillin ($100\,\mu g\,ml^{-1}$) as well as the cell permeability indicator PI ($1\,\mu g\,ml^{-1}$) and covered with a glass coverslip. Cells were imaged for 45 min with a 30 s frame rate and the number of PI-positive *E. coli* cells was counted from twenty 30 × 30 µm fields of view with at least 50% confluence of cells.

**Fluorescence microscopy and photobleaching.** For fluorescence microscopy a Nikon Ti-E inverted motorized microscope with Perfect Focus System and Plan Apo 1003 Oil Ph3 DM (NA 1.4) objective lens was used. SPECTRA X light engine (Lumencore), ET-GFP (Chroma #49002) and ET-mCherry (Chroma #49008) filter set were used to excite and filter fluorescence. sCMOS camera pco.edge 4.2 (PCO, Germany; pixel size 65 nm) and VisiView software (Visitron Systems, Germany) were used to record images. Temperature was regulated at 30 °C and 95% humidity using an Okolab T-unit (Okolab). For high-speed image acquisition of sheath contraction, the field of view on the sCMOS camera was reduced to 400 lines and the image acquisition mode was set to streaming. This allowed image acquisition at a frame rate of 500 frames per second during a total of 5 s observation time.

For photobleaching experiments the GFP fluorescence was diminished using a VS-AOTF 488 nm Laser system mounted with iLas2 laser merge on the microscope, allowing simultaneous LED and laser illumination. Cells were monitored for sheath assembly for 2 min at a frame rate of 2 s per frame. After 30 s the laser was triggered with 100% output power for 0.1 ms per pixel and cells were photobleached along a line profile. Polymerizing sheaths were identified during image analysis and used for the generation of kymograms in Fiji. Polymerization speeds as well as fluorescence intensity profiles to identify 'bright', 'bleached' and 'dim' sections along the sheath were generated from kymograms. Distance from origin of sheath polymerization was measured after 30 s, 32 s, 20 s before end point of sheath assembly as well as the end point itself for each section. The end point of sheath assembly was identified as the last frame on which an increase of sheath length was clearly detected. Fluorescence intensity measurements were corrected for cytosolic background fluorescence at indicated time points.

**Statistical analysis.** Statistical parameters such as number of biological replicates and total analysed bacteria as well as levels of significance are reported in the figure legends. Linear regression analyses as well as one-way and two-way ANOVA with multiple comparisons and Tuckey *post hoc* test were calculated using GraphPad Prism version 6.05. If not indicated differently, data are represented as mean ± s.d.

**Data availability.** The authors declare that all data supporting the findings of this study are available from the corresponding authors upon request.

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

## Acknowledgements

We thank Mihai Ionescu for an excellent technical assistance in cloning and generating strains. The work was supported by SNSF Starting Grant BSSGI0_155778, SNSF grant 31003A_159525 and the University of Basel. A.V. was supported by the Biozentrum Basel International PhD Program 'Fellowships for Excellence'. HADA dye was a kind gift from Prof Dr Martin Thanbichler, University of Marburg, Germany.

## Author contributions

A.V., J.W. and M.B. designed experiments, analysed and interpreted the results. A.V. and J.W. acquired all experimental data. L.L. generated *tssM* and *tssJ V. cholerae* strains and performed their initial characterization. A.V. and M.B. wrote the manuscript, which was read and approved by all authors.

## Additional information

**Competing interests:** The authors declare no competing financial interests.

