## [Peer Review File · Nature Communications]

Reviewers' comments:

Reviewer #1 (Remarks to the Author):

Vettiger and colleagues report data of an interesting work related to the assembly of the T6SS contractile sheath in *Vibrio cholerae*. They conclusively demonstrate that they can generate large *Vibrio* spheroplasts with increased volume and sheath lengths, allowing to monitor sheath assembly. Using this tool, they (i) go further on specific points such as defining that sheaths contract in less than 2 ms (until this work, sheath contraction was estimated to occur in < 5ms) and (ii) provide new data such as the sheath polymerization mechanism. All together these data significantly increase our knowledge on the T6SS. I have a few minor concerns regarding this beautiful work.

1. It is not clear whether all *Vibrio* cells turn to form spheroplasts upon ampicillin treatment. May be showing large fields and a statistical analyses (number of rod and speheric cells in untreated and ampicillin-treated cultures) will allow readers to appreciate the efficiency of spheroplasts formation. If 100% of the cells are spheroplasted, the addition of such data will strengthen the conclusion that spheroplasts have conserved anti-bacterial activity (fig. 2e). The efficiency of killing by *Vibrio* spheroplasts should be compared to the killing activity of WT, rod-shaped, cells.
2. The sheath contraction was measured from sheaths up to 3 um long. However, they observed sheaths 8-um long (fig. S1c). It will be a significative addition if they can image contraction of such sheaths using 2-ms frames.
3. The discussion needs to be expanded.

Reviewer #2 (Remarks to the Author):

This is another report by the Basler's lab on the mechanism by which the bacterial type VI secretion system (T6SS) operates. This time the main message is to suggest that sheath elongation occurs from the distal end. This is not per se such a surprising finding while most of the statements are based on a single approach which, although interesting and original, introduces a large number of biases which do not allow making definitive conclusions.

More specific comments are as below:

- The authors have a results section explaining that sheath length is limited by the cell size (lines 61-80). They tried to show that the T6SS remains active in *Vibrio cholerae* cells treated with ampicillin (which led to formation of spheroplasts). Monitoring the T6SS is exclusively performed using fluorescence microscopy. Indeed, in Fig 1A, S1A and video S1 and S2, elongated foci corresponding to sheath structures in spheroplasts are described. Yet, the presence of sheath tubes in cells is not a "clue" of an active T6SS. In several studies, it has been shown that sheath structures can be observed in cells lacking essential components of the T6SS (Kapitein et al. 2013; Planamente et al. 2016) or in the absence of T6SS. For example, recombinant production of sheaths proteins from *V. cholerae* and *P. aeruginosa* in *E. coli* cells spontaneously leads to tube formation (Bonemann et al. 2009; Lossi et al. 2013). Moreover, cycles of assembly-elongation, contraction and disassembly steps (in this strict order) are rarely observed in the imaged spheroplasts where the majority of sheaths display aberrant dynamics (as showed in video S1 and S2). Overall, the observed fluorescent data are not particularly convincing and the use of other approaches is required to show that the T6SS is active in the treated cells.

- In another results section, the authors suggest that spheroplasts assemble functional T6SS (lines 82-105). This could be interesting but partly redundant with the previous section. Furthermore,

and this is likely one major issue in this work, the authors should address the consequences of spheroplast formation on the integrity of the cell envelope. The problem here will concern the assembly/stability of the TssJLM complex which has been described as required for T6SS assembly (Durand et al. 2015). If this is true, how do the observed sheath tubes properly assembled in spheroplasts? This would need to be discussed.

- Another part in the results section aims at concluding that sheath contraction is faster than the temporal resolution limit (lines 107-117). Here, the authors attempted to measure sheath contraction and corresponding speed. For this, a total of 50 sheaths have been observed by fluorescence microscopy and one can query whether such a number is sufficient for a robust analysis. Furthermore, contracted sheaths are determined based on (i) a reduced sheath length (about 50 %) and (ii) an increase in thickness, which is in agreement with our current knowledge about sheaths from contractile injection machines (T6SS, T4 phage, pyocins...). However, several biases are introduced here. At this resolution and with the sole observation of fluorescent signals, it is difficult to assess sheath dimensions as only the fluorescent signal can be "measured". Discriminating if the "shorter and thicker" foci correspond to a contracted sheath is also questionable. For example, the observed foci might be the result of polysheaths or not properly assembled sheaths. It is unclear whether the selected sheaths used for these measurements are displaying proper dynamics. For example, in the panel 3 of Figure S2, the sheath seems to be in the middle of the spheroplast and does not seem to elongate near-perpendicular to the "membrane". How can such type of measurements can be deemed reliable?

Finally the key section of the results suggest that the sheath polymerizes at the distal end (lines 119-174). I do have a hard time to consider that the data presented are convincing enough. In brief, the bleaching data showed in Figure 4A are for the least peculiar. The authors attempt to observe sheath assembly/elongation in spheroplasts by bleaching a chosen section of the sheath and observe subsequent fate of the fluorescence (Figure 4A). A graphical representation of the observed fluorescence is sketched in Figure 4B. It seems to me that more than one single cell/spheroplast are present in the field. Therefore, the observed fluorescence might correspond to different events of sheath formation as these foci can potentially be seen in different cells. The authors drew a single cell (in Figure 4B) and represent what they believe is happening, which to me sounds more subjective than strictly rigorous. Why are the corresponding bright field images not shown?

- Finally, the whole study is poorly discussed and some of the statements in the discussion are out of context. For example, the authors state, lines 181-182, that their study provides new clues for the molecular mechanisms by which TssA proteins colocalizing to the distal end of an assembling sheath may promote sheath polymerization. There are not a single experiment that introduce TssA in this study, so this is unclear to me how such a major conclusion can be proposed. I am not even sure why a cap protein would be needed, considering that the authors propose the idea that the sheath subunits are added at the distal end and that the sheath length is regulated by the available space and the amount of subunits.

Minor comments:

- Based on observation about the length of the sheaths in *V. cholerae* spheroplasts versus the ones observed in the untreated cells, the authors conclude that "the length of T6SS sheath is limited by available space". This conclusion is in agreement with the observed data (Figure 1 and S1). However, an observation mentioned further in the manuscript (see lines 149-150) is slightly contradicting with this: the amount of the sheath subunits produced (VipA and VipB produced in cell) can be also a limitation factor of the sheath length. The authors did not really discuss these contradictions.
- In the c panel of Figure S1, I do not have clear clue of what I am looking at. Are these smashed spheroplasts? Merged spheroplasts?
- In addition to the approaches based on their previous Cell paper, using again fluorescence microscopy, which is nice and exciting but far too indirect, the authors may be willing to introduce

other approaches such as simple secretion assay. All necessary controls would need to be included.

- Legends and nomenclatures shown in several figures are confusing.

Reviewer #1 (Remarks to the Author):

Vettiger and colleagues report data of an interesting work related to the assembly of the T6SS contractile sheath in *Vibrio cholerae*. They conclusively demonstrate that they can generate large *Vibrio* spheroplasts with increased volume and sheath lengths, allowing to monitor sheath assembly. Using this tool, they (i) go further on specific points such as defining that sheaths contract in less than 2 ms (until this work, sheath contraction was estimated to occur in < 5ms) and (ii) provide new data such as the sheath polymerization mechanism. All together these data significantly increase our knowledge on the T6SS. I have a few minor concerns regarding this beautiful work.

We would like to thank the reviewer for the nice comments about our work.

1. It is not clear whether all *Vibrio* cells turn to form spheroplasts upon ampicillin treatment. May be showing large fields and a statistical analyses (number of rod and speheric cells in untreated and ampicillin-treated cultures) will allow readers to appreciate the efficiency of spheroplasts formation. If 100% of the cells are spheroplasted, the addition of such data will strengthen the conclusion that spheroplasts have conserved anti-bacterial activity (fig. 2e). The efficiency of killing by *Vibrio* spheroplasts should be compared to the killing activity of WT, rod-shaped, cells.

We have now included Figures 1b and S1b, which show the quantification of spheroplast conversion in time and large fields of view containing in total about 400 cells treated by ampicillin for 20 min and 40 min. We have repeated this experiment three times and detected more than 90% of cells converting to spheroplasts while maintaining active sheath dynamics after 40 min incubation in the presence of ampicillin. Importantly, there were less than 3% rod shaped cells with active T6SS. In addition, we have also included an *E. coli* permeability assay, which directly compares activity of rod-shaped cells and spheroplasts (Fig. 2e). As described below, we have also included comparison of Hcp secretion by spheroplasts and rod-shaped cells.

However, when we reanalyzed the *E. coli* killing assay performed with spheroplasts on agar plate, we realized that we recover only about 1% of spheroplasts added to the competition after 2 h of co-incubation with *E. coli*. This is likely due to lysis of spheroplasts on the agar plate as no decrease of *V. cholerae* recovery was observed for rod-shaped cells. This extensive lysis of spheroplasts likely explains why in this assay spheroplasts kill *E. coli* less efficiently than untreated cells. Spheroplast lysis complicates the interpretation of the *E. coli* killing assay, therefore, we decided to remove this experiment from the revised manuscript. Importantly, since we tested the T6SS activity in spheroplasts using three independent assays (Hcp secretion, *E. coli* permeabilization under cover slip, and VgrG2

exchange), we are convinced that our conclusion that T6SS remains active in spheroplasts is well supported by the data.

2. The sheath contraction was measured from sheaths up to 3 μm long. However, they observed sheaths 8- μm long (fig. S1c). It will be a significant addition if they can image contraction of such sheaths using 2-ms frames.

We agree with the reviewer that it would be interesting to also image much longer sheaths. However, one should take some technical limitations into consideration. To get longer sheaths, we need larger spheroplasts, which then have too many overlapping structures and this makes it difficult to resolve them individually. To overcome this problem, we can introduce the mutations that decrease assembly initiation frequency, however, this inevitably decreases the frequency of contractions. To image at a 2 ms frame rate, we have to limit the field of view on our camera to 16% and also significantly increase excitation intensity to collect enough light. This means that we only have a very short time window (about 5 s) to detect contractions before the structures photobleach to an extent which prevents reliable detection of changes in their length. The contractions happen stochastically and we cannot predict which structures contract at what time. Therefore, we blindly collect large number of image series, from a limited area, and then search for the contraction events. To obtain the data shown in this publication, we repeated the experiments more than 20 times and collected hundreds of GB of images. To perform this analysis with significantly longer sheaths, we would need to increase the number of collected image series by at least an order of magnitude.

Importantly, there is no indication that increasing the time resolution (by analyzing longer sheaths) by a factor of 2 or 3 would help to time resolve the contraction event. Therefore, we think that with current limitation of the available live-cell imaging technology, we are unable to measure how quickly sheaths contract. One should realize that to determine the speed of contraction, we would need to obtain at least 4 consecutive images of a sheath undergoing contraction (one fully extended, 2 with different levels of contraction and another image with fully contracted sheath). So far, we failed to detect even a single image between the extended and contracted state. Since the speed of contraction could be easily an order of magnitude faster than the limit we detected now, it is very likely that an entirely different approach will have to be developed in the future.

3. The discussion needs to be expanded.

Also in the light of the comments from Review #2, the discussion is now significantly expanded.

Reviewer #2 (Remarks to the Author):

This is another report by the Basler's lab on the mechanism by which the bacterial type VI secretion system (T6SS) operates. This time the main message is to suggest that sheath elongation occurs from the distal end. This is not per se such a surprising finding while most of the statements are based on a single approach which, although interesting and original, introduces a large number of biases which do not allow making definitive conclusions.

We are not aware of any previous studies addressing the issue of polarity of sheath polymerization, certainly not in T6SS field. If we omitted some direct evidence obtained in the past, we would be more than happy to discuss the data. To unambiguously identify the polarity of sheath assembly under native conditions one needs an oriented spatial and temporal observation of the assembly inside cells. We consider direct imaging of the T6SS to be an ideal approach for addressing this problem. We are unaware of any other approach, which would allow us to reach an unequivocal conclusion about polarity of sheath assembly in live cells.

More specific comments are as below:

- The authors have a results section explaining that sheath length is limited by the cell size (lines 61-80). They tried to show that the T6SS remains active in *Vibrio cholerae* cells treated with ampicillin (which led to formation of spheroplasts). Monitoring the T6SS is exclusively performed using fluorescence microscopy. Indeed, in Fig 1A, S1A and video S1 and S2, elongated foci corresponding to sheath structures in spheroplasts are described. Yet, the presence of sheath tubes in cells is not a "clue" of an active T6SS. In several studies, it has been shown that sheath structures can be observed in cells lacking essential components of the T6SS (Kapitein et al. 2013; Planamente et al. 2016) or in the absence of T6SS. For example, recombinant production of sheaths proteins from *V. cholerae* and *P. aeruginosa* in *E. coli* cells spontaneously leads to tube formation (Bonemann et al. 2009; Lossi et al. 2013). Moreover, cycles of assembly-elongation, contraction and disassembly steps (in this strict order) are rarely observed in the imaged spheroplasts where the majority of sheaths display aberrant dynamics (as showed in video S1 and S2). Overall, the observed fluorescent data are not particularly convincing and the use of other approaches is required to show that the T6SS is active in the treated cells.

In the manuscript, we now provide three different assays that show that T6SS remains active in the ampicillin treated cells. We show that: (i) T6SS positive spheroplasts secrete Hcp into culture supernatant to the same extent as untreated rod-shaped cells (Figures 2a, S2a); (ii) *E. coli* cells are permeabilized in a T6SS-dependent manner during co-incubation with *V. cholerae* spheroplasts (Fig. 2d, e, S2c); (iii) spheroplasts deliver VgrG2 protein into neighboring cells (Fig. 2b, c, S2b).

We agree that the sheath dynamics was difficult to follow in spheroplasts at the frame rate of the videos we have originally included in our manuscript. To demonstrate that sheaths in spheroplasts are indeed undergoing the expected cycle of assembly, contraction and disassembly by ClpV, we now include

further panels in Fig. 1c, 1d and 4a. These figures show sheath assembly into an extended structure spanning across the entire cell, which subsequently contracts and is being disassembled. In addition, we provide a new Video S1 showing the full time course of spheroplast induction. These movies show that cells, which displayed T6SS activity (dynamic VipA and ClpV signals) as rod-shaped cells maintain the T6SS activity throughout spheroplast formation. Occasionally, formation of static VipA-msfGFP spots are observed during spheroplast induction, however, those are clearly distinguishable from dynamic T6SS sheaths.

We agree with the reviewer that making sure that no polysheaths are formed is critical and that simply seeing the spots or structures of VipA-GFP is not a definitive evidence of T6SS function. Indeed, non-dynamic spots of VipA-GFP may appear in cells without functional T6SS as reported previously (Basler and Mekalanos, 2012; Basler et al., 2012; Kapitein et al., 2013). However, it is important to point out that polysheaths can only form when ClpV function is impaired because the polysheaths are structurally similar to contracted sheaths (as was shown for both T6SS and phage sheaths) and are therefore disassembled by ClpV (Bönemann et al., 2009). We show that in the spheroplasts, the ClpV is active because it localizes to contracted sheaths just after their contraction and this is followed by sheath disassembly. The ClpV dynamics in the spheroplasts is identical to that of rod-shaped cells shown in this study or previously (Basler and Mekalanos, 2012). Additionally, we show that frequency of sheath assembly in spheroplasts depends on the presence of VgrG1 and VasX to the same extent as in rod-shaped cells (Figure 4b), further confirming that these dynamic sheaths assemble from baseplates unlike polysheaths.

- In another results section, the authors suggest that spheroplasts assemble functional T6SS (lines 82-105). This could be interesting but partly redundant with the previous section. Furthermore, and this is likely one major issue in this work, the authors should address the consequences of spheroplast formation on the integrity of the cell envelope. The problem here will concern the assembly/stability of the TssJLM complex which has been described as required for T6SS assembly (Durand et al. 2015). If this is true, how do the observed sheath tubes properly assemble in spheroplasts? This would need to be discussed.

As was shown previously (Dörr et al., 2015) and as we show here as well, the cellular membranes remain intact during spheroplast formation. Only upon prolonged incubation times (> 60 min, 500 µg/ml ampicillin) examples of plasmolysis could be detected (Figure S1a - outer membrane is highlighted by arrow heads). Therefore, we do not see any reason why T6SS could not be active in spheroplasts with an intact outer membrane, which stays in contact with the inner membrane. The simplest explanation for the observation that T6SS is indeed active in the spheroplasts is that the membrane complex forms without attachment to peptidoglycan. Additionally, as shown recently, a transglycosylase (MltE) is specifically recruited to the point of T6SS assembly and peptidoglycan cleavage is essential for T6SS assembly (Santin and Cascales, 2016). These points are now discussed in lines 209 – 216.

- Another part in the results section aims at concluding that sheath contraction is faster than the temporal resolution limit (lines 107-117). Here, the authors attempted to measure sheath contraction and corresponding speed. For this, a total of 50 sheaths have been observed by fluorescence microscopy and one can query whether such a number is sufficient for a robust analysis. Furthermore, contracted sheaths are determined based on (i) a reduced sheath length (about 50 %) and (ii) an increase in thickness, which is in agreement with our current knowledge about sheaths from contractile injection machines (T6SS, T4 phage, pyocins...). However, several biases are introduced here. At this resolution and with the sole observation of fluorescent signals, it is difficult to assess sheath dimensions as only the fluorescent signal can be “measured”. Discriminating if the “shorter and thicker” foci correspond to a contracted sheath is also questionable. For example, the observed foci might be the result of polysheaths or not properly assembled sheaths. It is unclear whether the selected sheaths used for these measurements are displaying proper dynamics. For example, in the panel 3 of Figure S2, the sheath seems to be in the middle of the spheroplast and does not seem to elongate near-perpendicular to the “membrane”. How can such type of measurements can be deemed reliable?

We apologize if the text describing the contraction measurements was not clear enough. All the sheaths that were analyzed for the measurements of speed or level of contraction clearly transitioned from extended conformation to contracted conformation between two consecutive frames. This is now explained in the text (lines 121 - 124). We want to also point out that we use the sheath length measurement as the method to identify sheath contractions. We do not simply look for “thinner” and “thicker” sheaths as we agree with the reviewer that it is indeed impossible to measure thickness of the sheaths by light microscopy. The fact that contracted sheaths appear “thicker” is the consequence of their higher brightness due to the contraction and thus increased density of the GFP along the structure. However, sheaths are long enough to measure their length by diffraction limited fluorescence microscopy. The longer the sheaths are the lower the relative error of this measurement is. That was the reason why we reassessed the previous measurements in spheroplast with long sheaths. We want to stress that all the individual measurement points presented in Fig. 3c were based on analysis of the same sheath, which clearly contracted in a short time (between two subsequent frames) and the lengths before and after were compared as reported in the Fig. S2. We are convinced that such analysis is possible if the time period between the two length measurements is short. As was reported previously, after contraction the sheaths are not immediately disassembled, it takes about 10-20s to observe sheath disassembly. This is also clear from the new Video S1-S3 and Figure 1c provided to address the points raised above.

Finally the key section of the results suggest that the sheath polymerizes at the distal end (lines 119-174). I do have a hard time to consider that the data presented are convincing enough. In brief, the bleaching data showed in Figure 4A are for the least peculiar. The authors attempt to observe sheath assembly/elongation in spheroplasts by bleaching a chosen section of the sheath and observe subsequent fate of the fluorescence (Figure 4A). A graphical representation of the observed fluorescence is sketched in Figure 4B. It seems to me that more than one single cell/spheroplast are present in the field. Therefore, the observed fluorescence might correspond to different events of sheath formation as these foci can potentially be seen in different cells. The authors drew a single cell

(in Figure 4B) and represent what they believe is happening, which to me sounds more subjective than strictly rigorous. Why are the corresponding bright field images not shown?

To identify the polymerizing sheath before and after photobleaching we look at three critical parameters: (i) speed of assembly, (ii) direction of assembly, and (iii) location of the structure. For all depicted examples, we generated kymograms and measured speed of assembly before and after photobleaching. This clearly showed that the sheaths after photo-bleaching keep assembling with unaltered speed and direction. Kymogram of a non-bleached sheath is now shown in Fig. 4d for comparison. We also provide an additional example of polymerization of a photobleached sheath (Figure 5a, S6). Bright field images are now shown in Fig. 5a to make it clear where the cell boundaries are.

It is extremely unlikely that a new sheath assembly would start at the exact place and time where the previous sheath was photo-bleached. Furthermore, this new sheath would have to keep assembling in the exact same direction and with the same speed. As can be seen from the images, frequency of sheath initiation is relatively low under those imaging conditions and our analysis shows that polymerization speeds may vary among different assemblies. It is also important to stress that we never observe an assembling sheath structure simply jumping from one place to another between individual frames.

- Finally, the whole study is poorly discussed and some of the statements in the discussion are out of context. For example, the authors state, lines 181-182, that their study provides new clues for the molecular mechanisms by which TssA proteins colocalizing to the distal end of an assembling sheath may promote sheath polymerization. There are not a single experiment that introduce TssA in this study, so this is unclear to me how such a major conclusion can be proposed. I am not even sure why a cap protein would be needed, considering that the authors propose the idea that the sheath subunits are added at the distal end and that the sheath length is regulated by the available space and the amount of subunits.

We have now completely rewritten and expanded the discussion where we explain why the observation that sheath assembles at the end distal from the baseplate is important for considering the role of certain TssA proteins.

Minor comments:

- Based on observation about the length of the sheaths in *V. cholerae* spheroplasts versus the ones observed in the untreated cells, the authors conclude that “the length of T6SS sheath is limited by available space”. This conclusion is in agreement with the observed data (Figure 1 and S1). However, an observation mentioned further in the manuscript (see lines 149-150) is slightly contradicting with this: the amount of the sheath subunits produced (VipA and VipB produced in cell) can be also a limitation factor of the sheath length. The authors did not really discuss these contradictions.

We should have been more specific in the statements. This is now corrected in lines 217 - 223. These points are also discussed in the modified discussion.

- In the c panel of Figure S1, I do not have clear clue of what I am looking at. Are these smashed spheroplasts? Merged spheroplasts?

Those are single spheroplasts, incubated for a longer period of time, as labeled in the figure and the legend.

- In addition to the approaches based on their previous Cell paper, using again fluorescence microscopy, which is nice and exciting but far too indirect, the authors may be willing to introduce other approaches such as simple secretion assay. All necessary controls would need to be included.

We have discussed this issue above and introduced the suggested secretion assay in Fig. 2a. It is important to point out that simple Hcp secretion assay is less significant for assessing T6SS functionality than assays that specifically measure delivery of proteins into target cells. A potential problem with spheroplasts is that they lyse more readily than rod-shaped cells and thus release Hcp into the supernatant. Nevertheless, we minimized this issue by washing the spheroplasts prior to the inoculation into fresh medium and by following Hcp secretion only for a limited amount of time. Importantly, the spheroplasts indeed secrete Hcp as efficiently as rod-shaped cells (Fig. 2a).

- Legends and nomenclatures shown in several figures are confusing.

We have made small edits to both figures and legends. We would be happy to fix anything that is still confusing or incomplete.

Reviewers' comments:

Reviewer #1 (Remarks to the Author):

The authors have appropriately addressed my comments, both experimentally (Hcp secretion assay) and by editing the discussion. The manuscript has been nicely improved. I do certainly understand the experimental limitations that the authors have noticed regarding imaging sheath contraction in very large spheroplasts.

Reviewer #2 (Remarks to the Author):

I have reviewed this manuscript previously and already pointed at a number of issues. The authors have addressed several of these points although they do not always provide direct answers.

In essence I had two main issues:

1. The first issue was to wonder whether assembly of the T6SS sheath from the distal end is surprising and a major breakthrough. This is what one would expect when making a direct comparison with the events that may occur in other contractile sheaths such as with bacteriophages. The authors argue that it is novel and has never been shown in the T6SS, which I fully agree and am fully aware of but I was simply questioning whether it represents a T6SS breakthrough. Previous studies, including from the Basler's lab, have provided atomic details of T6SS sheaths (Kudryashev et al. 2015; Clemens et al. 2015) in which it appears that T6SS and T4 phage sheaths share a conserved assembly.
2. The second point I am struggling with is the use of spheroplasts and all the interpretation which is made from the data generated. I fully acknowledge the elegance and beauty of the approach and images provided, the cleverness of the authors and all the cutting edge methodologies used, but in the end one should make sure that the set up used can answer the question without any ambiguity or biases. I would no longer argue on the basis of the images or new kymograms provided whether what we look at are genuine events and no such things like polysheaths. The authors seem convinced it is not and discuss it extensively. Nevertheless, key data they indicate they provide to demonstrate that the T6SS is functional in the spheroplasts are very much puzzling. The secretion assay in figure 2a for example shows that there is Hcp secretion in a VipB-dependent manner and indeed there are no traces of Hcp in the supernatant of vipB spheroplasts. Yet the authors should provide an additional control with an intracellular protein to show that there is no leakage. The authors could argue that if Hcp is out in the WT and not in the vipB mutant this is good a demonstration. However, in their rebuttal they make clear that spheroplasts are lysing easily so it is unclear why the vipB spheroplasts are not leaking any Hcp produced. The other points this reviewer made earlier is the fact that the T6SS spans the cell envelope and spheroplasting might disrupt the structural integrity of this complex. The authors argue it is not. If there is no T6SS disruption then using a tssJ mutant (instead of a vipB mutant) should show that in this case as well no Hcp secretion is observed and would fully demonstrate the membrane complex is required even in the spheroplast. I suggest TssJ since it is the outer membrane component but any other components like TssLM or a putative baseplate component would help addressing this issue. It should also be possible to co-localize TssM with the sheath in spheroplasts since in previous papers TssM can be labelled and foci are detected if the docking platform is assembled (Durand et al. 2015 or Santin and Cascales 2016). Addressing few of these points will definitely make the study very solid and reassuring as for the interpretation proposed.

REVIEWERS' COMMENTS for the revised manuscript:

Reviewer #1 (Remarks to the Author):

The authors have appropriately addressed my comments, both experimentally (Hcp secretion assay) and by editing the discussion. The manuscript has been nicely improved. I do certainly understand the experimental limitations that the authors have noticed regarding imaging sheath contraction in very large spheroplasts.

We would like to thank the reviewer for reading the revised version of the manuscript and for the nice words about our work.

Reviewer #2 (Remarks to the Author):

I have reviewed this manuscript previously and already pointed at a number of issues. The authors have addressed several of these points although they do not always provide direct answers.

We thank the reviewer for commenting on the revised version of our manuscript. As explained below, the requested experiments are now included in the final revision of the manuscript.

In essence I had two main issues:

1. The first issue was to wonder whether assembly of the T6SS sheath from the distal end is surprising and a major breakthrough. This is what one would expect when making a direct comparison with the events that may occur in other contractile sheaths such as with bacteriophages. The authors argue that it is novel and has never been shown in the T6SS, which I fully agree and am fully aware of but I was simply questioning whether it represents a T6SS breakthrough. Previous studies, including from the Basler's lab, have provided atomic details of T6SS sheaths (Kudryashev et al. 2015; Clemens et al. 2015) in which it appears that T6SS and T4 phage sheaths share a conserved assembly.

2. The second point I am struggling with is the use of spheroplasts and all the interpretation which is made from the data generated. I fully acknowledge the elegance and beauty of the approach and images provided, the cleverness of the authors and all the cutting edge methodologies used, but in the end one should make sure that the set up used can answer the question without any ambiguity or biases. I would no longer argue on the basis of the images or new kymograms provided whether what we look at are genuine events and no such things like polysheaths. The authors seem convinced it is not

and discuss it extensively. Nevertheless, key data they indicate they provide to demonstrate that the T6SS is functional in the spheroplasts are very much puzzling. The secretion assay in figure 2a for example shows that there is Hcp secretion in a VipB-dependent manner and indeed there are no traces of Hcp in the supernatant of *vipB* spheroplasts. Yet the authors should provide an additional control with an intracellular protein to show that there is no leakage. The authors could argue that if Hcp is out in the WT and not in the *vipB* mutant this is good a demonstration. However, in their rebuttal they make clear that spheroplasts are lysing easily so it is unclear why the *vipB* spheroplasts are not leaking any Hcp produced. The other points this reviewer made earlier is the fact that the T6SS spans the cell envelope and spheroplasting might disrupt the structural integrity of this complex. The authors argue it is not. If there is no T6SS disruption then using a *tssJ* mutant (instead of a *vipB* mutant) should show that in this case as well no Hcp secretion is observed and would fully demonstrate the membrane complex is required even in the spheroplast. I suggest *TssJ* since it is the outer membrane component but any other components like *TssLM* or a putative baseplate component would help addressing this issue. It should also be possible to co-localize *TssM* with the sheath in spheroplasts since in previous papers *TssM* can be labelled and foci are detected if the docking platform is assembled (Durand et al. 2015 or Santin and Cascales 2016). Addressing few of these points will definitely make the study very solid and reassuring as for the interpretation proposed.

We respectfully disagree with the reviewer that additional lysis control is necessary for the Hcp secretion assay. Hcp is an abundant cytosolic protein and the fact that it is undetectable in the supernatant of *vipB* deletion strain clearly shows that no significant lysis of the spheroplasts occurs during this experiment. Additionally, we want to point out that we provide Coomassie stained gel of the supernatant samples. As can be seen in the Supplementary figure S2a, the pattern and amount of the proteins detected in all samples is similar. This clearly shows that there is no major difference between the compared samples and therefore no significant lysis occurs due to ampicillin treatment during this particular experiment. However, we do not find it surprising that during more elaborate experiments, lysis of spheroplasts may occur. For example, recovery of cells from an agar plate after several hours of competition with *E. coli* requires extensive manipulation of the sample (vortexing and resuspending by pipetting), which the spheroplasts may not be able to survive as we pointed out previously.

Following the reviewer's suggestion, we tested whether *TssJ* and *TssM* are required for sheath assembly and Hcp secretion in spheroplasts. As shown in Figure 2, deletion of *tssJ* or *tssM* reduces sheath assembly in spheroplasts to the same level as in the untreated cells. Moreover, both *tssJ* and *tssM* negative cells or spheroplasts are unable to secrete Hcp (Figure 2a, S2a). These additional controls are now described in the revised text, lines 92-93 and 217-218.

We are convinced that this, together with all the other experimental evidence discussed in the manuscript, clearly establishes that spheroplasts assemble active T6SS that secretes Hcp and delivers proteins to the neighboring cells. Therefore, the observation that sheath subunits assemble at the sheath end distal from the membrane anchor is generally valid.